

# Exploring the mediating roles of physical literacy and mindfulness on psychological distress and life satisfaction among college students

Wencong Kan[1], Fan Huang[2], Menglin Xu[3], Xiangyun Shi[4], Zengyin Yan[5] and Mehmet Türegün[6]

[1] Department of Sports Teaching and Research, Lanzhou University, Lanzhou, China
[2] University of California, San Diego, La Jolla, CA, United States of America
[3] Ohio State University, Columbus, OH, United States of America
[4] Nanjing University of Science and Technology, Nanjing, China
[5] Chongqing University of Posts and Telecommunications, Chongqing, China
[6] Barry University, Miami, United States of America

## ABSTRACT

**Background**. Psychological distress has been a growing challenge to healthy living worldwide. Special attention has been concentrated on examining the cost of psychological distress on the life satisfaction of college students who are vulnerable groups coping with the challenge. The purpose of this study is to explore the roles of physical literacy (PL) and mindfulness in mediating the impact of psychological distress on life satisfaction among college students in China.

**Methods**. A sample of 653 students from six universities across three cities in China participated in an online survey, which included measures of PL, mindfulness, life satisfaction, as well as stress, anxiety, and depression levels. Structural equation modeling (SEM) was implemented to analyze the survey data.

**Results**. The findings of the SEM analysis demonstrated an acceptable model fit ($X^2/df$ = 3.63, CFI = 0.951, TLI = 0.940, RMSEA = 0.068, 90% CI = [0.060, 0.075], SRMR = 0.051) with a large effect size ($R^2$ = 0.36) for life satisfaction, indicating that 36% of the variation in life satisfaction could be explained by the model. In addition, significant partial-mediation effects of PL and mindfulness were observed in the relationship between psychological distress and life satisfaction. These findings provide empirical support for the notion that interventions targeting PL and mindfulness practices may effectively enhance well-being and alleviate psychological distress among college students. Furthermore, this study suggests that integrating PL and mindfulness components into physical education and activity programs could be beneficial in meeting individuals' holistic health needs.

# INTRODUCTION

Psychological distress has increasingly become a significant threat to global well-being. Among vulnerable population, college students face unique challenges during their

Corresponding author
Fan Huang, hefu411@126.com

transition to adulthood, including academic pressures, social communication, and financial issues (*Acharya, Jin & Collins, 2018*; *Soysa & Wilcomb, 2015*). Chinese college students, in particular, had reported experiencing psychological distress, for example, stress, anxiety, and depression, during their academic journey (*Wu et al., 2015*; *Zhang et al., 2018*). These negative feelings can significantly impact students' subjective well-being, encompassing their life satisfaction (*Dong et al., 2020*; *Ryff et al., 2006*). Subjective well-being, theorized by *Diener & Chan (2011)*, involves individuals' psychological evaluations of their lives in relation to positive and negative experiences. A model of subjective well-being has been proposed which consists of three components, namely, pleasant affect, unpleasant affect, and life satisfaction (*Diener, RE & Smith, 1999*). Physical activity (PA) has been widely recognized for its positive effects in alleviating psychological distress (*Biddle & Asare, 2011*; *Tyson et al., 2010*). However, the prevalence of physical inactivity (PI), which is inversely associated with mental health (*Galper et al., 2006*), has become a dominant lifestyle gradually being increased across societies. *Dumith et al. (2011)* found that PI was prevalent in both less developed and highly developed countries, with one-fifth of adults failing to meet the minimum PA levels of optimal health. Similarly, *Wu et al. (2015)* reported low engagement in high-level PA among Chinese college students, with only 7.7% participating regularly. The escalating prevalence of PI presents a serious global health challenge, contributing to various diseases and adversely affecting public health (*Lee et al., 2012*). Addressing this issue is crucial for promoting overall well-being and mitigating the negative impact of psychological distress on college students and society.

Physical literacy (PL) has recently gained prominence as a central goal in physical education and PA programs worldwide, aiming to enhance public health outcomes (*Castelli, Barcelona & Bryant, 2015*; *Choi et al., 2018*; *Young et al., 2020*). PL is a comprehensive educational concept, rooted in the philosophical traditions of phenomenology, existentialism, and monism (*Whitehead, 2001*; *Young et al., 2020*). According to the International Physical Literacy Association (*International Physical Literacy Association, 2017*), "*Physical literacy can be described as the motivation, confidence, physical competence, knowledge and understanding to value and take responsibility for engagement in physical activity for life*" (*International Physical Literacy Association, 2017*). PL is conceptualized across four domains: affective, physical competence, cognitive, and behavioral (*Caldwell et al., 2020*; *Tremblay et al., 2018*). Enhancing PL not only fosters engagement in PA but also protects against exercise-related injuries and illness resulting from PI (*Jurbala, 2015*). Studies have shown a positive association between PL and PA engagement (*Caldwell et al., 2020*; *Choi et al., 2018*; *Dong et al., 2023*; *Kwan et al., 2019*; *Ma et al., 2020*). For example, *Kwan et al. (2019)* reported that a PL-based intervention increased first-year college students' positive attitudes toward PA, correlating with higher engagement in PA. *Ma et al. (2020)* found a positive association between PL and PA among college students, while *Choi et al. (2018)* reported a similar relationship among adolescents aged 12 to 18 years. While promoting PA is a primary benefit of PL, its advantages extend beyond physical health. *Bremer, Graham & Cairney (2020)* conceptualized that PL contributes to psychological and social well-being by encouraging active participation in PA. *Caldwell et al. (2020)* investigated the relationship between PL and health indicators in school-aged

children, revealing positive correlations with PA and health-related quality of life, along with negative correlations with blood pressure and body fat percentage. These findings present the diverse benefits of PL, emphasizing its potential to enhance overall well-being beyond physical health.

Mindfulness serves as a valuable strategy for decreasing psychological distress while promoting subjective well-being. Initially termed consciousness discipline, mindfulness was conceptualized by *Walsh (1980)* as achieving inner peace and freedom through specific mental training to enhance perception and consciousness of emotions and experiences. *Kabat-Zinn (2003)* later rooted mindfulness in Buddhist philosophy, defining it as the non-judgmental awareness of moment-to-moment experiences, including feelings, emotions, and body sensations. Mindfulness encourages individuals to adopt a neutral stance in observing their thoughts and experiences, foresting acceptance of the present moment (*Baer, 2003*). Failure to cultivate mindfulness may lead to negative outcomes, as *Hanson & Mendius (2009)* discovered that individuals often experience distress from their reactions to neutral events, projecting their own needs and desires onto situations. *Segal, Williams & Teasdale (2002)* noted that a goal-oriented lifestyle may hinder acceptance of reality, leading to worry about the future and rumination on past events. In contrast, mindfulness promotes a being-oriented lifestyle, emphasizing acceptance and experiencing the present moment as it is (*Segal, Williams & Teasdale, 2002*). By disengaging from automatic thought patterns, individuals can recognize that thoughts are not facts, thereby reducing stereotypical and prejudiced thinking (*Baer, 2003*; *Langer & Moldoveanu, 2000*). Shifting to a mindful perspective enables individuals to reduce negative emotions, such as stress, depression, and anxiety. Studies have investigated the relationship between mindfulness and well-being and psychological distress (*e.g.,* *Bamber & Morpeth, 2019*; *Bajaj & Pande, 2016*; *Dvořáková et al., 2017*; *McDonald et al., 2016*; *Ramler et al., 2016*). For instance, *Dvořáková et al. (2017)* found that mindfulness training improved life satisfaction and reduced depression and anxiety among college students in the United States. Similarly, *Bajaj & Pande (2016)* observed a positive relationship between mindfulness, resilience, and life satisfaction among college students in India. A meta-analytic study by *Bamber & Morpeth (2019)* reported that mindfulness-based interventions decreased anxiety among college students, with moderate to large effect sizes. *McDonald et al. (2016)* illustrated a similar negative relationship between mindfulness and psychological distress in Australia. Despite these findings, it remains unclear whether mindfulness mediates the relationship between psychological distress and life satisfaction. Therefore, further research investigating the mediating role of mindfulness in this relationship is warranted.

Theories surrounding both PL and mindfulness have been posited to promote an individuals' overall well-being while alleviating psychological distress (*Segal, Williams & Teasdale, 2002*; *Whitehead, 2010*). Moreover, mindfulness predominantly targets the mental domain, while aspects of the affective and cognitive domains of PL, such as taking reasonability for one's actions, could be reasonably construed as pertaining to the mental domain as well. While the health-promoting benefits of mindfulness and PL have been explored independently, there remains an unknown gap concerning whether mindfulness and the affective and cognitive domains of PL are related to each other and whether each

mediates the relationship between psychological distress and life satisfaction. Consequently, this study seeks to explore how PL and mindfulness contribute to mediating the effects of psychological distress on life satisfaction among university students in China. The findings could potentially advance our comprehension of the characteristics of PL and mindfulness. Drawing upon relevant theoretical frameworks, these research questions (RQ) and hypotheses have been formulated.

RQ 1: What are the relationships among psychological distress, life satisfaction, mindfulness, and PL?

RQ 2: Do PL and mindfulness mediate the relationship between life satisfaction and psychological distress?

$H_{1a}$: PL is negatively associated with psychological distress and positively associated with life satisfaction

$H_{1b}$: Mindfulness is negatively associated with psychological distress and positively associated with life satisfaction.

$H_2$: Mindfulness positively mediates the relationship between life satisfaction and psychological distress.

$H_3$: The relationship between life satisfaction and psychological distress is mediated by mindfulness.

## MATERIALS & METHODS

### Participants

A cross-sectional survey design was employed for this study. Convenience sampling was implemented to recruit participants from six universities located in three cities: Nanjing, Chongqing, and Xi'an, within mainland China. Five of these universities are public institutions situated in Nanjing, Chongqing, and Xi'an. Overall, 653 Chinese college students with 320 (49%) being female students participated in this study. The mean age of the participants was 19.20 (standard deviation [$SD$]= 1.31). Regarding the distribution of participants by location, 170 were from two universities in Chongqing, 28 were from one university in Xi'an, and 455 were from three universities in Nanjing. Participants represented a wide range of academic disciplines, including education, business, and computer science, among others.

### Measures
#### Satisfaction with Life Scale (SWLS)
The Satisfaction with Life Scale (SWLS; *Diener et al., 1985*) was the instrument adopted for measuring the levels of life satisfaction among participants in this research. The SWLS consists of five items, with responses rated on a 7-point Likert-type scale ranging from 1 (strongly disagree) to 7 (strongly agree), measuring global life satisfaction. The Cronbach's $\alpha$ coefficient has been reported as 0.92 for the overall scale, indicating excellent internal consistency reliability for the instrument, for the Chinese population (*Bai et al., 2011*).

#### Perceived Physical Literacy Instrument (PPLI)
The Perceived Physical Literacy Instrument (PPLI) was used to measure the levels of the affective and cognitive domains of PL among Chinese college students. The PPLI

consists of nine items distributed across three subscales: sense of self and self-confidence, self-expression and communication with others, and knowledge and understanding (*Sum et al., 2016*). Responses are recorded on a 5-point Likert scale ranging from strongly disagree to strongly agree. Specifically, the subscales of sense of self and self-confidence and self-expression and communication with others measure individuals' motivation and confidence while participating in PA, representing the affective domains (*Sum et al., 2018*). Examples of items include ''I possess self-evaluation skills for health'', ''I am confident in wild/natural survival'', ''I am capable in handling problems and difficulties'', and ''I possess self-management skills for fitness'' among others (*Sum et al., 2016*; *Sum et al., 2018*). Moreover, the subscale of knowledge and understanding evaluates participants' awareness of the benefit of PA, representing the cognitive domain of PL. Examples of items in the subscale include ''I appreciate myself or others doing sports'' and ''I am aware of the benefits of sports related to health'', among others (*Sum et al., 2016*; *Sum et al., 2018*). Each subscale displayed a good internal consistency, with reported Cronbach's $\alpha$ coefficients of 0.73 for a sense of self and self-confidence; 0.76 for self-expression and communication with others; and 0.76 for knowledge and understanding among the Chinese population (*Sum et al., 2016*). It is worth noting that the PPLI instrument has been used for measuring PL across different age groups within the Chinese population, with consistent findings of good internal consistency (*Choi et al., 2021*; *Li et al., 2021*; *Sum et al., 2018*). Given the study's focus on investigating the effects of the affective and cognitive domains of PL on psychological distress and life satisfaction, the physical competence and behavioral domains of PL were not included in the analysis. We acquired permission to use this instrument from the author.

### Depression anxiety stress scale—21 items (DASS-21)

The Depression Anxiety Stress Scale—21 Items (DASS-21) is a short version of the Depression Anxiety Stress Scale—42 Items (DASS-42; *Lovibond & Lovibond, 1995*). Based on the original DASS-21, the Chinese version of DASS-21 consists of three subscales of depression, anxiety, and stress. Each subscale has seven items describing negative emotional or physical responses for assessing the depression, anxiety, and stress of the past week using a 4-point Likert-type scale from 0 (did not apply to me at all) to 3 (applied to me very much) (*Gong et al., 2010*). The instrument indicated good internal consistency reliability for these subscales as Cronbach's $\alpha$ coefficients for the Chinese version of DASS were reported as 0.77 for the depression subscale; 0.79 for the anxiety subscale; 0.76 for the stress subscale; and 0.89 for the total scale-based on an investigation using 1779 Chinese college students from three different universities in Beijing (*Gong et al., 2010*). The DASS is in the public domain and no special permission is needed for research or clinical purposes.

## The Mindful Attention Awareness Scale (MAAS)

The state MAAS consists of five items with a 6-point Likert-type response scale from 1 to 6 (almost always to almost never) for assessing individuals' attention to and awareness of what is happening in the present (*Brown & Ryan, 2003*). The state MAAS indicated an excellent internal consistency Cronbach's $\alpha$ coefficients were reported as 0.92 (*Brown*

& *Ryan, 2003*). An example of an item includes "I was doing something without paying attention". The state MAAS instrument was adopted for this study to measure college students' mindfulness. The trait MAAS is in the public domain and special permission is not required to use it for research purposes.

To ensure the validity of the translations, we translated the surveys, consulted bilingual experts, and incorporated their suggestions through revisions.

## Procedures

Following approval from the Barry University's Institution Review Board (Protocol Number: 1651073), convenience sampling was employed to recruit participants from various regions in China. Collaboration was sought from six faculty members employed by Chinese universities, who were contacted *via* email and requested to assist in participant recruitment. These faculty members represented universities located in different cities in China: one in Xi'an, two in Chongqing, and three in Nanjing. All six contacted faculty members agreed to participate in collecting data. Each faculty member received a recruitment email containing a link to the online survey and was asked to forward this link to potential participants. Upon accessing the online survey *via* the provided link, participants were presented with an informed consent form on the first page. Participants who consented to participate clicked a "Yes" button on the informed consent form before proceeding to complete the online survey.

## Data analysis

The goal of the study was to examine the mediation effects of PL and mindfulness on psychological distress and life satisfaction. First, descriptive statistics were used to summarize responses to the survey items, including means, *SD*, kurtosis, skewness, and Pearson correlation. In addition, Cronbach's $\alpha$ coefficient was provided for each subscale using the R package *psych* (*Revelle, 2021*). Second, structural equation modeling (SEM), a sophisticated technique for investigating a theoretical-based model with goodness-of-fit indices, was applied. SEM is a powerful tool for hypothesis testing and theory development because it allows researchers to test complex models that incorporate multiple variables and relationships simultaneously. SEM was implemented to assess the relationships among the independent variable, dependent variable, and mediators. Specifically, a multiple-mediator analysis was conducted to investigate the hypotheses regarding the affective and cognitive domains of PL and mindfulness as mediators between life satisfaction and psychological distress. To address the concern about violating the normality assumption, the maximum likelihood estimator with robust (MLR) (*Rosseel, 2020*) was used for examining the multiple-mediator analysis with the four latent variables (LVs). Moreover, psychological distress was loaded by sum scores of each subscale of stress, anxiety, and depression; PL was loaded by sum scores of each subscale of sense of self and sense of confidence, self-expression, and communication with others, and knowledge and understanding; and life satisfaction and mindfulness were loaded by five manifest variables (MVs) each for their scales. In addition, it is important to note that the PL consisted of two domains: affective and cognitive.

The goodness-of-fit indices for assessing the model fit included $X^2/degree\ of\ freedom\ (df)$ ratio, Comparative Fit Index (CFI), Tucker Lewis Index (TLI), Standardized Root Mean Square Residual (SRMR), and Root Mean Square Error of Approximation (RMSEA), suggested by *Hu & Bentler (1999)*. The criteria of the goodness-of-fit were as follows: a good model fit: TLI >0.95, CFI >0.95, SRMR <0.06, and RMSEA <0.06, and an acceptable model fit: TLI >0.90, CFI >0.90, SRMR <0.08, and RMSEA <0.08 (*Hu & Bentler, 1999*; *Schumacker & Lomax, 2010*). Moreover, a bootstrapping of 5,000 samples of original data was used to test the significance of the two mediation effects. This study used the R package *lavaan*, developed by *Rosseel (2012)*, to conduct the SEM models.

## RESULTS

### Summary statistics

Cronbach's $\alpha$ coefficient was conducted to examine internal consistency reliability for each subscale of the instruments. The range of Cronbach's alpha coefficients across all study instruments and subscales was from 0.72 to 0.89, indicating good internal consistency reliability for each subscale. Moreover, the first hypothesis was confirmed, demonstrating that PL and mindfulness exhibit negative associations with psychological distress and positive associations with life satisfaction. Pearson's correlation analysis was used to assess the correlations between each MV. As shown in Table 1, the MVs of life satisfaction, the affective and cognitive domains of PL, and mindfulness were positively correlated while these MVs were negatively correlated with psychological distress loaded by stress, depression, and anxiety. The strongest association as displayed in Table 1 was 0.83 between ls2 and ls3, was from the same scale of life satisfaction. There are concerns about a possible violation of the normality assumption as some of the skewness values were close to 1.

### Structural equation modeling evaluation

A multiple-mediator analysis was conducted to examine the mediation effects of PL and mindfulness on the relationship between psychological distress and life satisfaction. The second and third hypotheses were validated, indicating that PL mediates the relationship between life satisfaction and psychological distress, and this relationship is also mediated by mindfulness. The results of the structural model indicated an acceptable model fit ($X^2/df = 3.63$, CFI = 0.951, TLI = 0.940, RMSEA = 0.068, 90% confidence interval [CI] = [0.060, 0.075], SRMR = 0.051). Moreover, the MVs were statistically significantly loaded ($p < 0.001$) to their respective LVs. In addition, the relationship between two IVs of PL and mindfulness was statistically significant and positive correlated (unstandardized coefficient [b] = 0.337, standard error [SE] = 0.070, $t = 4.804$, $p < 0.001$). The structural model was comprised of five statistically significant and direct path coefficients and two statistically significant and indirect path coefficients. The LV of psychological distress was statistically significantly and negatively associated with the LVs of life satisfaction ($b = -0.061$, $SE = 0.021$, $t = -2.867$, $p < 0.01$), PL ($b = -0.286$, $SE = 0.038$, $t = -7.488$, $p < 0.001$), and mindfulness ($b = -0.191$, $SE = 0.016$, $t = -11.913$, $p < 0.001$). In contrast, the LV of life satisfaction was statistically significant and positively associated with the LVs of PL ($b = 0.213$, $SE = 0.026$, $t = 8.192$, $p < 0.001$) and mindfulness ($b = 0.176$, $SE =$

Kan et al. (2024), *PeerJ*, DOI 10.7717/peerj.17741

Peer J

**Table 1 Descriptive statistics and correlation matrix.**

| | ls1 | ls2 | ls3 | ls4 | ls5 | anxt | dept | strt | Kut | ssct | set | mf1 | mf2 | mf3 | mf4 | mf5 |
|---|---|---|---|---|---|---|---|---|---|---|---|---|---|---|---|---|
| ls1 | – | | | | | | | | | | | | | | | |
| ls2 | 0.78*** | – | | | | | | | | | | | | | | |
| ls3 | 0.77*** | 0.83*** | – | | | | | | | | | | | | | |
| ls4 | 0.57*** | 0.58*** | 0.63*** | – | | | | | | | | | | | | |
| ls5 | 0.47*** | 0.50*** | 0.51*** | 0.50*** | – | | | | | | | | | | | |
| anxt | −0.28*** | −0.32*** | −0.29*** | −0.25*** | −0.21*** | – | | | | | | | | | | |
| dept | −0.35*** | −0.34*** | −0.35*** | −0.23*** | −0.20*** | 0.74*** | – | | | | | | | | | |
| strt | −0.31*** | −0.34*** | −0.30*** | −0.23*** | −0.26*** | 0.79*** | 0.75*** | – | | | | | | | | |
| kut | 0.32*** | 0.37*** | 0.34*** | 0.25*** | 0.13*** | −0.22*** | −0.28*** | −0.17*** | – | | | | | | | |
| ssct | 0.38*** | 0.41*** | 0.36*** | 0.33*** | 0.27*** | −0.23*** | −0.23*** | −0.23*** | 0.60*** | – | | | | | | |
| set | 0.39*** | 0.45*** | 0.43*** | 0.37*** | 0.34*** | −0.26*** | −0.29*** | −0.30*** | 0.45*** | 0.64*** | – | | | | | |
| mf1 | 0.23*** | 0.29*** | 0.29*** | 0.20*** | 0.23*** | −0.39*** | −0.39*** | −0.37*** | 0.17*** | 0.21*** | 0.25*** | – | | | | |
| mf2 | 0.27*** | 0.30*** | 0.30*** | 0.21*** | 0.23*** | −0.37*** | −0.39*** | −0.39*** | 0.15*** | 0.22*** | 0.30*** | 0.69*** | – | | | |
| mf3 | 0.16*** | 0.19*** | 0.18*** | 0.18*** | 0.20*** | −0.31*** | −0.28*** | −0.31*** | 0.05 | 0.19*** | 0.19*** | 0.42*** | 0.50*** | – | | |
| mf4 | 0.22*** | 0.27*** | 0.28*** | 0.18*** | 0.15*** | −0.42*** | −0.44*** | −0.42*** | 0.21*** | 0.26*** | 0.29*** | 0.51*** | 0.56*** | 0.44*** | – | |
| mf5 | 0.27*** | 0.28*** | 0.31*** | 0.26*** | 0.23*** | −0.41*** | −0.43*** | −0.41*** | 0.23*** | 0.28*** | 0.29*** | 0.50*** | 0.54*** | 0.44*** | 0.70*** | – |
| *M* | 4.78 | 4.76 | 4.81 | 4.58 | 3.74 | 4.15 | 3.30 | 5.17 | 12.30 | 10.03 | 9.90 | 4.21 | 4.08 | 3.73 | 4.19 | 4.23 |
| *SD* | 1.19 | 1.19 | 1.21 | 1.30 | 1.75 | 3.04 | 3.22 | 3.61 | 1.99 | 2.46 | 2.20 | 1.16 | 1.10 | 1.15 | 1.20 | 1.17 |
| *skew.* | −0.06 | 0.04 | −0.07 | 0.14 | 0.18 | 0.75 | 0.97 | 0.43 | −0.24 | 0.06 | 0.05 | −0.35 | −0.17 | 0.32 | −0.22 | −0.19 |
| *kurt* | −0.05 | −0.22 | −0.24 | −0.58 | −0.84 | −0.04 | 0.14 | −0.46 | −0.81 | −0.36 | −0.13 | −0.22 | −0.46 | −0.74 | −0.65 | −0.73 |
| *α* | 0.89 | | | | | 0.79 | 0.83 | 0.80 | 0.77 | 0.80 | 0.72 | 0.85 | | | | |

**Notes.**

*** $p < 0.001$

ls, life satisfaction; anxt, anxiety; dept, depression; strt, stress; kut, knowledge and understanding; ssct, sense of self and sense of confidence; set, self-expression and communication with others; mf, mindfulness; skew, skewness; kurt, kurtosis; $\alpha$, Cronbach's alpha.

The same hereinafter

**Table 2  Structural model path coefficients.**

| LVs | b | β | 95% CI | | SE | t |
|---|---|---|---|---|---|---|
| pd–ls | −0.061 | −0.162 | −0.102 | −0.019 | 0.021 | −2.867** |
| pd–pl | −0.286 | −0.365 | −0.361 | −0.211 | 0.038 | −7.488*** |
| pd–mf | −0.191 | −0.605 | −0.223 | −0.160 | 0.016 | −11.913*** |
| pl–ls | 0.213 | 0.447 | 0.162 | 0.264 | 0.026 | 8.192*** |
| mf–ls | 0.176 | 0.149 | 0.054 | 0.297 | 0.062 | 2.836** |

Notes.
*$p < 0.05$
**$p < 0.01$
***$p < 0.001$

pd, psychological distress; pl, physical literacy; ls, life satisfaction; mf, mindfulness; b, unstandardized coefficient; β, standardized coefficient; 95% CI, 95% confidence interval; SE, standard error.
The same hereinafter.

0.062, $t = 2.836$, $p < 0.01$), as displayed in Table 2 and Fig. 1. Furthermore, $R^2$ value for life satisfaction was 0.36 indicating that 36% of the variation in life satisfaction could be explained by the structural model; the $R^2$ value for mindfulness was 0.366 indicating that 36.6% of the variation in mindfulness could be explained by the structural model; and the $R^2$ value for PL was 0.133 indicating that 13.3% of the variation in PL could be explained by the structural model. In addition, $R^2$ values for life satisfaction and mindfulness yielded a large effect size and the $R^2$ value for PL showed a medium effect size (*Cohen, 1988*).

Two statistically significant partial-mediation effects were found. Psychological distress had a statistically significant and indirect effect on life satisfaction mediated by PL ($b = -0.061$, 95% CI = [−0.084, −0.038], $SE = 0.012$, $t = -5.240$, $p < 0.001$). Similarly, psychological distress had a statistically significant and indirect effect on life satisfaction mediated by mindfulness ($b = -0.034$, 95% CI = [−0.057, −0.010], $SE = 0.012$, $t = -2.750$, $p < 0.01$). Moreover, a total effect, the sum of the two indirect effects and one direct path of psychological distress on the life satisfaction, was found to be statistically significant ($b = -0.155$, 95% CI = [−0.191, −0.119], $SE = 0.018$, $t = -8.480$, $p < 0.001$). Further, a non-parametric bootstrap method was used to test the significance of the two mediation effects. The results of a bootstrapping of 5,000 samples of original data showed that mediation effects of PL ($b = -0.061$, 95% CI = [−0.086, −0.040], $SE = 0.012$, $t = -5.128$, $p < 0.001$) and mindfulness ($b = -0.034$, 95% CI = [−0.058, −0.010], $SE = 0.012$, $t = -2.722$, $p < 0.01$) were significant because the standardized path coefficient 95% CI did not include 0.

## DISCUSSION

The present study examined the simultaneous relationships among LVs of PL, mindfulness, psychological distress, and life satisfaction. The structural multiple-mediator model indicated an acceptable model fit, revealing five significant direct effects and two significant indirect effects. Additionally, 36% of the variation in life satisfaction, 36.6% of the variation in mindfulness, and 13.3% of the variation in PL could be explained by the structural multiple-mediator model simultaneously.

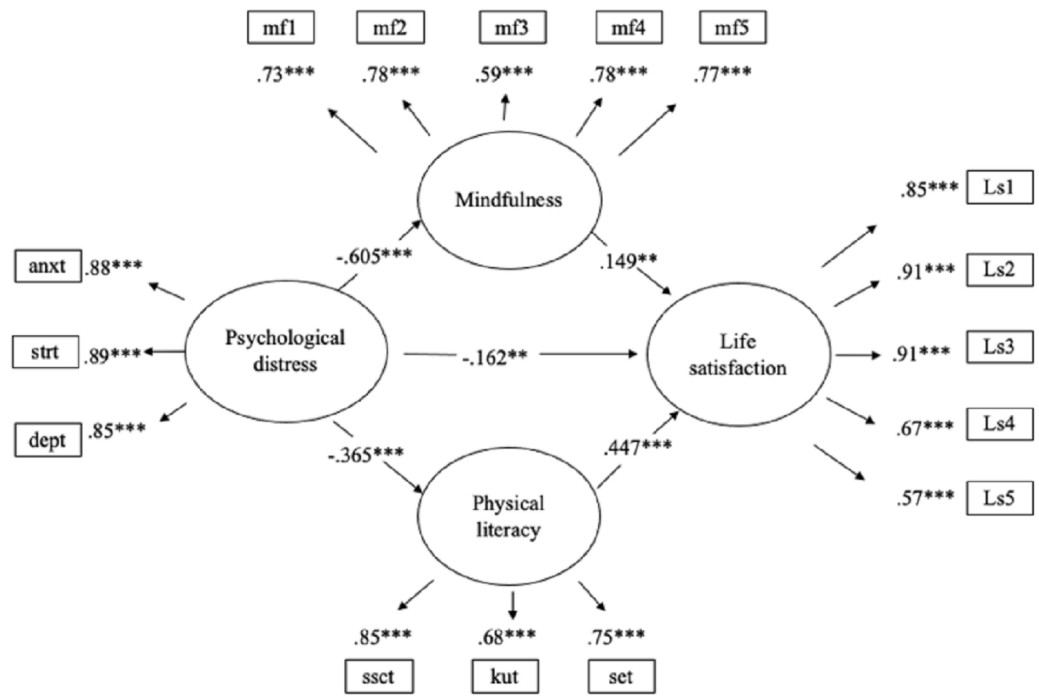

**Figure 1** **The standardized coefficients of mediation SEM.** Note. * *p* < 0.05, ** *p* < 0.01, and *** *p* < 0.001. ls, life satisfaction; anxt, anxiety; dept, depression; strt, stress; kut, knowledge and understanding; ssct, sense of self and sense of confidence; set, self-expression and communication with others; mf, mindfulness.

The results of the study support the three hypotheses that drove the study. First, the present results indicated that the LV of PL loaded by knowledge and understanding, sense of self and sense of confidence, and self-expression and communication with others, which represented the affective and cognitive domains of PL, were negatively associated with the LV of psychological distress loaded by stress, anxiety, and depression. The finding suggests that individuals with higher levels of PL experience lower psychological distress. This finding is consistent with theories proposed by *Bremer, Graham & Cairney (2020)* and *Castelli, Barcelona & Bryant (2015)*, which underscore the association between PL and social and psychological aspects. Essentially, the findings highlight that the affective and cognitive domains of PL are inversely related to psychological distress, including stress, anxiety, and depression, among college students.

Second, the results indicated that the LV of PL loaded by knowledge and understanding, sense of self and sense of confidence, and self-expression and communication with others was statistically significantly and positively associated with the LV of life satisfaction. This suggests that individuals who experienced a higher level of PL had a higher level of life satisfaction. These results could be complemented and enriched by considering the statistical evidence from the study conducted by *Caldwell et al. (2020)*, which found associations between PL and various health indicators, including health-related quality of life, body composition, physical fitness, and blood pressure. Moreover, the present results

were consistent with the PL theories conceptualized by *Jurbala (2015)* and *Whitehead (2010)*, suggesting that individuals with positive attitudes and profound knowledge and understanding of the value of PL tend to have a higher degree of subjective well-being compared to those with lower levels of positivity and understanding. Therefore, it is reasonable to expect that PL plays a role in promoting a higher level of life satisfaction among individuals.

Third, the results indicated that both PL and mindfulness were mediators in the relationship between life satisfaction and psychological distress. Moreover, the results illustrated a positive correlation between PL and mindfulness, highlighting that both constructs could be construed as belonging to the mental domain. Additionally, the results also showed that the affective and cognitive domains of PL and mindfulness mediated the relationship between psychological distress loaded by stress, anxiety, depression, and life satisfaction simultaneously. The present results are consistent with the study conducted by *Bajaj & Pande (2016)*, which reported a positive correlation between mindfulness and life satisfaction, along with a negative correlation with negative affect. The present findings provide further support and suggest potential generalizability beyond the specific population studies. Furthermore, the results of the study revealed that mindfulness played a mediator in the relationship between life satisfaction and psychological distress loaded by stress, anxiety, and depression. The findings align with theories of PL suggesting that its affective and cognitive domains of PL can promote positive attitudes and mitigate psychological challenges associated with maintaining a healthy lifestyle (*Cairney et al., 2019*). The statistical evidence may support a reciprocal pathway as conceptualized by *Caldwell et al. (2020)*, wherein promoting PL leads to increased participation in PA, ultimately improving mental health across all age groups within a population.

These findings suggest that PL can serve as a critical element of physical education programs aimed at encouraging a lifelong healthy lifestyle (*Castelli, Barcelona & Bryant, 2015*). From this perspective, the present results suggest that the affective and cognitive domains of PL may play a mediating role in diminishing stress, anxiety, and depression while improving individuals' well-being. This interpretation aligns with the concept of integrating body and mind, as proposed by *Whitehead (2001)*. Conceptually, PL not only fosters increased engagement in PA, but it also addresses individuals' psychological well-being simultaneously. This concept resonates with the theory of monism. As articulated by *Whitehead (2001)* and *Whitehead (2010)*. Moreover, the present findings indicate that the mediation effects of PL and mindfulness can complement each other in promoting individuals' sense of well-being. This implies that interventions aimed at enhancing life satisfaction could benefit from integrating mindfulness and PL strategies. Such an integrated approach may offer a more holistic framework for improving individuals' overall quality of life and mental health.

## LIMITATION

Since this study used a cross-sectional survey design, it is essential to acknowledge its limitation in establishing causality. Even though the study found statistical evidence

suggesting significant associations among the LVs of PL, mindfulness, psychological distress, and life satisfaction, the findings cannot conclude causal relationships among these LVs. Therefore, future studies are needed with experimental investigations examining potential causality among the variables of PL, mindfulness, psychological distress, and subjective well-being among the college student population. Moreover, the present study employed a convenience sampling technique to recruit participants, potentially restricting the generalizability of the results to the entire student population. Future studies should include a more diverse and representative sample. Future studies may examine the relationships among the variables to different student populations to enhance the generalizability of the findings. Furthermore, although the instruments used in this study were previously validated for Chinese students (*Bai et al., 2011*; *Dong et al., 2020*; *Dong et al., 2023*; *Gong et al., 2010*), cultural factors might have influenced the results. Future studies could investigate the cultural validation of these instruments. Additionally, this study focused on individuals' perceived PL, specifically targeting aspects of knowledge and understanding, sense of self and confidence, and self-expression and communication with others, representing the affective and cognitive domains of PL. Future studies should explore the impacts of physical competence and behavioral domains of PL on subjective well-being within the college student population, contributing to a deeper understanding of PL characteristics.

## CONCLUSION

This study represents a pioneering empirical investigation into the relationships among the LVs of the affective and cognitive domains of PL, mindfulness, psychological distress, and life satisfaction among college students. The present results of the SEM indicated an acceptable model fit, highlighting the potential explanatory power of PL, mindfulness, and psychological distress in understanding individuals' life satisfaction. The findings offer empirical support for the notion that theories of PL and mindfulness hold promise in fostering well-being and mitigating psychological distress among college students. Moreover, the study implies that interventions aimed at promoting PL and mindfulness practices could enhance the effectiveness of physical education and activity programs, fulfilling to individuals' holistic health needs.

## ACKNOWLEDGEMENTS

We would like to thank Prof. Gerene K. Starratt for providing valuable suggestions.

### Funding
The authors received no funding for this work.

### Competing Interests
The authors declare there are no competing interests.

## Author Contributions

- Wencong Kan conceived and designed the experiments, analyzed the data, prepared figures and/or tables, authored or reviewed drafts of the article, and approved the final draft.
- Fan Huang conceived and designed the experiments, authored or reviewed drafts of the article, and approved the final draft.
- Menglin Xu conceived and designed the experiments, authored or reviewed drafts of the article, and approved the final draft.
- Xiangyun Shi performed the experiments, authored or reviewed drafts of the article, and approved the final draft.
- Zengyin Yan performed the experiments, authored or reviewed drafts of the article, and approved the final draft.
- Mehmet Türegün conceived and designed the experiments, authored or reviewed drafts of the article, and approved the final draft.

## Human Ethics

The following information was supplied relating to ethical approvals (*i.e.*, approving body and any reference numbers):

Barry University

## Data Availability

The raw measurements are available in the Supplementary File.

## Supplemental Information

Supplemental information for this article can be found online at http://dx.doi.org/10.7717/peerj.17741#supplemental-information.

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
