# Peer review of "Exploring the mediating roles of physical literacy and mindfulness on psychological distress and life satisfaction among college students"

_PeerJ, doi:10.7717/peerj.17741_

## Round 0.1 · original submission · Major Revisions

The study addresses important issues related to the mental health and well-being of university students, employing Structural Equation Modeling (SEM) appropriately for the complex relationships between variables and using well-established scales (SWLS, PPLI, DASS-21, MAAS) to ensure high-quality data. However, I recommend improving the formulation of hypotheses to follow the hypothetico-deductive method more clearly. The research should clearly identify the central question, formulate specific hypotheses, and establish clear predictions derived from these hypotheses. Results should be aligned with the hypotheses, stating clearly whether each hypothesis was accepted or refuted, avoiding statistical jargon.Additionally, the study has limitations (please, provide a limitation topic): there is insufficient detail on how the translations of the instruments were culturally validated, which could affect the validity of the results, and the use of a convenience sample limits the generalizability of the findings to the entire population of university students, potentially introducing biases. Despite these limitations, the study is a valuable contribution to understanding the interactions between physical literacy, mindfulness, psychological distress, and life satisfaction, and addressing these recommendations would further strengthen the research.

Reviewer 1 ·

Basic reporting

The manuscript is written in professional English with technical terms appropriately used, facilitating understanding of complex concepts. The comprehensive literature review cites studies that establish the relationship between psychological distress and interventions like physical literacy and mindfulness. The article is well-organized with a logical flow. Tables and figures are effectively integrated, presenting data clearly. If possible, provide access to raw data to support transparency and reproducibility. The results section directly addresses and supports the hypotheses posed at the beginning of the paper. Consider discussing any limitations in more detail that might affect the generalizability of the results, such as the use of convenience sampling and the cultural context which might influence the applicability of the results to other populations.

Experimental design

The study on physical literacy and mindfulness pertains to health and psychological well-being, aligning with the journal’s aims. The research question is well articulated and addresses a meaningful gap in existing literature concerning the effects of physical literacy and mindfulness on psychological distress among college students. Using structural equation modeling provides a robust framework for analyzing complex relationships, indicating a high technical standard. The manuscript describes the instruments and analytical techniques used, facilitating understanding of the methods employed.

Validity of the findings

The study employs a strong statistical methodology, and the manuscript discusses control measures that guarantee the integrity of the data analysis. The conclusions are directly tied to the research hypotheses. I would suggest replication by specifying how future studies could build on these results, perhaps by applying the study’s methodology to different demographic groups or integrating additional variables.

Reviewer 2 ·

Basic reporting

Clear English. There are sufficient references. The structure of the paper is very good. The results are connected with the hypotheses.

Experimental design

The methodology is well explained and supported.

Validity of the findings

No comments.

Additional comments

The paper is well structured. The literary review is very good. The resuslts are indeed a contribution to the topic.

Annotated reviews are not available for download in order to protect the identity of reviewers who chose to remain anonymous.

Reviewer 3 ·

Basic reporting

1. The manuscript generally appears well-written with professional English, providing clarity on the methodology, results, and conclusions. Ensure that technical terms and acronyms (e.g., SEM, PL) are defined when first introduced to make the paper accessible to readers unfamiliar with specific jargon.
2. The paper references relevant literature, discussing the impact of psychological distress on college students and the potential benefits of physical literacy and mindfulness.
3. The structure of the article is professional, with logically organized sections and clear headings. Tables and figures are used effectively to summarize data and illustrate findings. Consider adding more descriptive captions for figures and tables to explain what they depict and how they relate to the text.
4. The paper does a good job of presenting results that directly address the stated hypotheses. The findings are relevant and clearly tied back to the hypotheses in the discussion.

Experimental design

1. The manuscript aligns with the aims and scope of PeerJ, which includes publishing original research across biological, medical, and environmental sciences. The focus on physical literacy and mindfulness addresses relevant topics within health and psychology.
2. Research hypotheses are clear and address a significant gap in understanding the mediating roles of physical literacy and mindfulness in reducing psychological distress.
3. The choice of structural equation modeling (SEM) suggests a rigorous analytical approach, and adherence to ethical standards is indicated by the approval from the university's Institutional Review Board.
4. The methods section appears thorough, with detailed descriptions of the survey instruments and statistical techniques used. You could consider adding a supplementary file with the survey questionnaire and a more detailed protocol to enhance reproducibility.

Validity of the findings

1. The manuscript does not explicitly assess the novelty or potential impact of its findings within the broader field, which is crucial for understanding the study’s contribution to existing knowledge. Encourage meaningful replication by discussing how further studies could expand on your findings or apply your methodology to different populations or settings.
2. The manuscript provides comprehensive information on the data collection methods and statistical analysis, demonstrating thorough and controlled data management.
3. The conclusions are aligned with the research question, summarizing how the findings address the initial hypotheses regarding the mediating roles of physical literacy and mindfulness.

Additional comments

Authors could consider the readability of the manuscript for non-specialists by simplifying complex terminology and providing clear explanations of advanced concepts. This would make the paper more accessible to readers who are not experts in SEM or the specific psychological theories discussed.

---

## Round 0.2 · Minor Revisions

The structure of the presented research questions and hypotheses can be improved to ensure greater clarity and coherence. Firstly, research question RQ 1, which investigates the relationships among psychological distress, life satisfaction, mindfulness, and PL, could be formulated more specifically. The phrase "What are the relationships" can be replaced with "What are the relationships" for greater clarity.

The second research question, RQ 2, which asks if PL and mindfulness can mediate the relationship between life satisfaction and psychological distress, is clear but could be restructured to avoid redundancy. A suggestion would be: "Do PL and mindfulness mediate the relationship between life satisfaction and psychological distress?"

The presented hypotheses could also be improved. H1, which proposes that PL and mindfulness are negatively associated with psychological distress and positively associated with life satisfaction, is clear but could be divided into two for greater precision. For example, "H1a: PL is negatively associated with psychological distress and positively associated with life satisfaction" and "H1b: Mindfulness is negatively associated with psychological distress and positively associated with life satisfaction."

H2, which suggests that PL mediates the relationship between life satisfaction and psychological distress, is straightforward but could specify the direction of mediation. Thus, it could be reformulated as "PL positively mediates the relationship between life satisfaction and psychological distress." H3, which states that the relationship between life satisfaction and psychological distress is mediated by mindfulness, could also specify the direction of mediation, being reformulated to "Mindfulness positively mediates the relationship between life satisfaction and psychological distress."

The phrase "there research questions (RQ) and hypotheses have been formulated" contains a grammatical error ("there" instead of "these"), which should be corrected to "These research questions (RQ) and hypotheses have been formulated."

---

## Round 0.3 · accepted · Accept

Thank you for this improved version according to our comments.